# The Investigation of Giardiasis (Foodborne and Waterborne Diseases) in Buffaloes in Van Region, Türkiye: First Molecular Report of *Giardia duodenalis* Assemblage B from Buffaloes

**DOI:** 10.3390/pathogens12010106

**Published:** 2023-01-08

**Authors:** Özlem Orunç Kılınç, Adnan Ayan, Burçak Aslan Çelik, Özgür Yaşar Çelik, Nazmi Yüksek, Gürkan Akyıldız, Fatma Ertaş Oğuz

**Affiliations:** 1Özalp Vocational School, Van Yuzuncu Yil University, Van 65100, Turkey; 2Department of Genetics, Faculty of Veterinary Medicine, Van Yuzuncu Yil University, Van 65100, Turkey; 3Department of Parasitology, Faculty of Veterinary Medicine, Siirt University, Siirt 56100, Turkey; 4Department of Internal Medicine, Faculty of Veterinary Medicine, Siirt University, Siirt 56100, Turkey; 5Department of Internal Medicine, Faculty of Veterinary Medicine, Van Yuzuncu Yil University, Van 65100, Turkey; 6Department of Basic Health Sciences, Faculty of Health Sciences, Marmara University, Istanbul 34854, Turkey; 7Tuzluca Vocational School, Department of Medical Services and Techniques, Iğdır University, Iğdır 76000, Turkey

**Keywords:** assemblage B, buffalo, foodborne, *Giardia duodenalis*, first report

## Abstract

*Giardia duodenalis* (*G. duodenalis*) is an important zoonotic protozoan agent that causes foodborne and waterborne diarrhea in humans and other mammals. Molecular-based tests are critical in diagnosing giardiasis in humans and animals, identifying species, understanding the zoonotic potential and transmission routes, and evaluating taxonomy. Therefore, this study aimed to investigate the molecular characterization of *G. duodenalis* in buffaloes in the Van region in Türkiye. Buffaloes are a species that has been poorly studied in this regard. For this purpose, 100 fecal samples were collected from buffaloes in the Van region. The DNA extraction was performed using the GeneMATRIX STOOL DNA Purification Kit from stool samples. The nested PCR test was performed with the appropriate primers from the obtained DNA samples. The obtained bands suitable for sequencing were sent for sequence analysis, and the sequence results were aligned bidirectionally and compared with the database of GenBank by BLAST. As a result of the study, an 11% positivity rate for *G. duodenalis* was found in buffaloes, and assemblage E and assemblage B were isolated. To our knowledge, assemblage B in buffaloes was reported for the first time in this study. As a result, it was concluded that buffaloes are an important reservoir for waterborne and foodborne giardiasis.

## 1. Introduction

There are approximately 600 million cases of disease per year caused by bacteria, parasites, toxins, and chemicals from water and food, affecting close to 1 in 10 people worldwide. According to reports, 31 pathogens cause foodborne diseases, and 420 thousand people die annually from these diseases, with 125 thousand of these deaths being children under five years of age. Foodborne illnesses are more severe in countries with poor levels of development. The following conditions increase the incidence of these diseases: unsafe water in food production, preparation, or presentation; poor hygiene; and a low level of education. More than half of all foodborne illnesses are diarrheal diseases. Giardiasis is an important diarrheal disease caused by the protozoan *G. duodenalis* (syn. *Giardia intestinalis* and syn. *Giardia lamblia*), and the role of contaminated food in the transmission of giardiasis has been discussed previously [1,2,3,4]. In 2010, the World Health Organization (WHO) reported that *Giardia* caused 28.2 million cases of foodborne illness, and it has been included in the scope of the communicable diseases warning by the WHO. The United Nations Food and Agriculture Organization (FAO) and the WHO included giardiasis among the foodborne parasitic diseases in 2014. In giardiasis, infection occurs via the fecal–oral route. The transmission of *Giardia* infection occurs either by direct contact with the feces of infected mammals or by contaminated food, water, and beverages. The cyst form of *Giardia*, which plays a role in transmission and survives for months in water and soil, has been isolated from different foods (fruit, vegetables, and meat) [5,6,7,8]. As with other parasitic diseases, giardiasis has many more effects in immunocompromised or young hosts. Infection can be controlled in hosts with normal immunity. As little as 10 *Giardia* cysts can infect hosts who are vulnerable to infection. Therefore, contaminated water and nutrients with cysts can cause epidemics [2,9,10,11,12].

There are eight assemblages (A to H) of *G. duodenalis*, which can infect different mammalian species. The A and B assemblages responsible for human infection are isolated from dogs, cats, livestock, and wild animals. Therefore, *G. duodenalis* A and B assemblages are accepted as zoonoses. Molecular studies have shown that the main species responsible for giardiasis infection in farm animals are assemblage E, but there are also A and B assemblages. Furthermore, studies reported that assemblage E may also be present in humans. Ruminants are seen as the most important source in giardiasis epidemics because their feces production is very high [12,13,14,15,16]. Buffaloes are important ruminant species and are endangered. The presence of *G. duodenalis* A and E assemblages in buffalos has been reported, and they have been determined to be a reservoir host in terms of giardiasis, similar to other ruminants [17,18,19,20,21].

Understanding the epidemiology of foodborne diseases such as giardiasis and determining the zoonotic potential of its etiologic agent are essential for disease management, especially in areas of high prevalence in animals and/or humans. The investigation of the parasite in susceptible pet species that are suitable reservoirs for human infections will be of particular priority. Buffaloes are special animals that are more powerful than other ruminant animals, and their descendants are gradually decreasing. Studies on the prevalence of *Giardia* in buffaloes and the effects of buffaloes in transmitting zoonotic *Giardia* species are limited. Therefore, in this study, the prevalence of giardiasis in Anatolian Buffaloes (*Bubalis bubalis*), the *Giardia* genotypes in these animals, and whether the determined genotypes pose a danger to public health via the risk potential for foodborne diseases were investigated.

## 2. Materials and Methods

### 2.1. The Study Area

This study was carried out in the province of Van, located in the Eastern Anatolia Region of Türkiye in 2021 and 2022 (380 29′ N, 430 20′ E).

### 2.2. Animal Material and Sample Collection

For this study, stool samples were collected from the rectum of 100 Anatolian Buffaloes (*B. bubalis*) (0–4 years) in various enterprises in the Van region, with disposable latex gloves and stored in a stool container. The information on the buffalo were noted on the stool containers and brought to the laboratory on the same day.

### 2.3. Method

#### 2.3.1. Microscopic Analysis

In the microscopic examination, all 100 samples were examined under the microscope (Leica, Wetzlar, Germany) for *Giardia* cysts using the native Lugol method.

#### 2.3.2. DNA Extraction and Nested PCR

The collected samples were stored at −20 °C for DNA extraction. After 100 samples were completed, the DNA extraction was performed with a GeneMATRIX STOOL DNA Purification Kit from all samples. Nested PCR was performed from the extracted DNA. For this purpose, the β-giardin gene region was amplified using the primers (G7 F5′-AAGCCCGACGACCTCACCCGCAGTGC-3′ forward and G759R 5′-GAGGCCGCCCTGATCTTCGAGACGAC-3′ reverse) defined by Caccio et al. [21]. The following ingredients were in the 25 µL master mix: 10 picomoles of forward and reverse primers, 200 µM dNTPs, 1.5 mM MgCl2, 1 U Taq Polymerase, 10X PCR buffer (500 mM Tris-HCl, pH 8.8, 160 mM (NH4)SO_4_, and 0.1% Tween^®^20), nuclease-free water, and DNA. Following 15 min of predenaturation at 95 °C, the reaction took place with 35 cycles consisting of denaturation (30 s at 95 °C), annealing (30 s at 60 °C), elongation (1 min at 72 °C), and a final elongation of 7 min at 72 °C. The obtained PCR products were stored at +4 °C until the second PCR. The PCR products (BG1F 5′-GAACGAGATCGAGGTCCG-3′ forward and BG2R 5′-CTCGACGAGCTTCGTGTT-3′ reverse) were generated using the primers described by Lalle et al. [22]. For this purpose, 10 pmol of forward and reverse primers, 200 µM dNTPs, 1.5 mM MgCl2, 1 U Taq Polymerase, 10X PCR buffer (500 mM Tris-HCl, pH 8.8, 160 mM (NH4)SO4, and 0.1% Tween^®^20), nuclease-free water, and DNA were used in 25 µL of master mix. Following 15 min of predenaturation at 95 °C, the reaction took place with 35 cycles consisting of denaturation (30 s at 95 °C), annealing (30 s at 55 °C), elongation (1 min at 72 °C), and 7 min of final elongation at 72 °C. The PCR products obtained were stored at 4 °C until they were imaged in agarose gel. The PCR products were separated on a 1.5% agarose gel in an electrophoresis tank. The PCR products were run at a 90 volt linear current for approximately 60 min and photographed in the gel imaging device under UV light at the end of the period. The positive PCR products were sent to the sequence bilaterally. The PCR products were sequenced as forward and reverse. In this study, the PCR product that we obtained in our previous study, and which was confirmed by sequence analysis as *G. duodenalis* assemblage A, was used as a positive control [23].

#### 2.3.3. Sequence Analyzes

The DNA sequences obtained were checked individually in the BioEdit program, aligned, and made ready for analysis. The edited formats of the DNA sequences were compared with the dataset using the National Center for Biotechnology Information (NCBI) Basic Local Alignment Search Tool to determine the assemblages. In addition, the DNA data from KC960641, JQ978667, JQ978668, EU014384, HM165227, KC960635, DQ116616, HM165226, KP026313, KP026314, AY258618, AB714977, AY647264, AY072729, DQ116607, DQ116608, AY545647, AY545646, DQ090525, DQ090524, AY072725, AY072724, AY072723, AY545645, AY545644, and X85958 access-coded β-giardin gene sequences from the NCBI GenBank database were related to the set of files used and which samples were assembled. The reason why these access codes were preferred is that the primer sets we used in the PCR process were compatible with these regions. In addition, the assemblage B and E datasets were created according to the BLAST results of the study samples, and care was taken in that the datasets of the other assemblages included the sequence of the PCR product we produced. To create a phylogenetic tree, the data were aligned in the BioEdit program, and a model test was performed using the maximum likelihood statistical method in the Mega X program. Using the model determined from the model test, a phylogenetic tree of 1000 bootstraps was created using the Mega X program by the maximum likelihood statistical method (Appendix A).

## 3. Results

In this study, the *Giardia* screening was performed with the native Lugol and nested PCR methods in the feces of 100 buffaloes aged 0–4 years.

### 3.1. Native Lugol and Nested PCR Analysis Results

The *Giardia* cyst scanning was performed with the native Lugol examination method in 100 stool samples. This method detected *Giardia* cysts in eight stool samples (8%). Two animals identified as positive for *Giardia* had significant diarrhea. There was no significant change in the other feces. The *Giardia* cyst density was high in the diarrhea-consistent positive stools (see Figure 1). The hosts from which the diarrheal stools were taken were 0–6 months old. Molecular analyses were then performed with nested PCR from the DNA obtained from the stool samples. Then, *G. duodenalis*-specific bands of the expected size were obtained in 11 samples (11%). The appropriate bands for *G. duodenalis* were also obtained in the PCR test in all samples with *Giardia* cysts detected in the native Lugol examination. The number of examined samples, their age range, and positive sample numbers are given in Figure 2.

### 3.2. Sequence Analyses and NCBI Basic Local Alignment Search

Of the 11 positive samples, only four PCR products were found to be suitable for sequence analysis after comparing the DNA sequences of the β-giardin gene in the study with the database in the NCBI Basic Local Alignment Search Tool. An assemblage determination could not be made because the sequence data from seven samples were of poor quality. Of the samples suitable for sequence analysis, three belonged to the 0–6 month age group and one belonged to the 7–12 month age group. The comparison showed that 99.78% of Sample 1 of the sequence analysis was identical with (KT922248) assemblage E; 99.57% of Sample 2 was identical with assemblage E; 98.85% of Sample 3 was identical with (MG736344) assemblage B; and 99.76% of Sample 4 was identical with (MG736344) assemblage B (see Table 1). Since the model with the maximum likelihood value (InL) closest to zero was general-time reversible (gtr+G+I), this model was used when creating the phylogenetic tree. As in the results obtained from the NCBI GenBank database, in the phylogenetic tree created using the appropriate model, Samples 1 (OP985406) and 2 (OP985407) were located in the assemblage E clade, and Samples 3 (OP985408) and 4 (OP985409) were located in the assemblage B clade (see Figure 3). In this study, *G. duodenalis* assemblage B was found in buffaloes (*B. bubalis*) for the first time. The sequence results were uploaded to GenBank, and accession numbers were obtained. The comparison results of the study samples created using the NCBI Basic Local Alignment Search Tool are presented in Table 1.

According to the literature review, *G. duodenalis* assemblage B, which is significant in human giardiasis, has not been previously reported in buffaloes, appearing for the first time in our study. As seen in the phylogenetic tree created, assemblage E included Samples B and C, and Samples D and E were located in the assemblage B clade, specifically associated with human giardiasis. Sample D had a 98.85% overlap in the gene region related to the *G. duodenalis* assemblage B sample, with access code MG736244 isolated from human stool. The difference arises from the following five nucleotide changes: C-A in the 3rd position of the amplicon, C-G in the 16th position, C-G in the 58th position, C-G in the 142nd position, and A-G in the 343rd position. Of these changes, the change in position 3 causes a silent mutation. Mutations at other positions lead to some amino acid changes. Accordingly, mutations at positions 16 and 343 cause a change from arginine to glycine, and mutations at positions 58 and 142 cause a change from leucine to valine. Sample E also overlapped 99.79% in the gene region related to the *G. duodenalis* assemblage B sample, with access code MN457746 isolated from human stool. The difference is in the 267th amplicon, only due to the C-T variation, and this change is a silent mutation. Sample B from *G. duodenalis* assemblage E samples, isolated from buffaloes and associated with the ruminants we determined from this study, had a 99.78% overlap in terms of the gene region related to the *G. duodenalis* assemblage E sample, with access number GQ337972 isolated from sheep stool. The difference is due to a nucleotide A-G exchange at the 297th position of the amplicon, and this exchange causes a change from isoleucine to valine. Sample C had a 99.57% overlap in the gene region related to the *G. duodenalis* assemblage E sample, with access number KT922250 isolated from lamb stool. The difference is due to two nucleotide changes: A-G at position 103 and G-A at position 353 of the amplicon These changes are that do not cause any amino acid changes.

## 4. Discussion

*Giardia* infection (giardiasis) is one of the most common causes of water and foodborne diseases worldwide and has been neglected by the World Health Organization (WHO) [1,5]. Molecular-based tests are critical in diagnosing giardiasis in humans and animals, identifying species, understanding the zoonotic potential and transmission routes, and evaluating taxonomy. As a result of the increased molecular studies and discussions on giardiasis in recent years, the World Health Organization (WHO) has defined *G. duodenalis* as a zoonotic agent [5]. In previous studies, A and B assemblages isolated from humans were isolated from bovine species, and assemblage E, the prevalent genotype of bovids responsible for *Giardia* infection, was isolated from humans, which reveals that there are transmissions between ruminants and humans [13,22,23,24,25,26]. Buffalo is a crucial ruminant species, and studies investigating the prevalence and genotypes of giardiasis in these animals are insufficient. In studies on buffaloes around the world, a prevalence of 1.9% was found in Australia [27], 6.56% in Brazil [28], 19.8% in Italy [29], and 20% in Iraq [30]. In this study in Türkiye, the prevalence was determined as 11%. Studies have been conducted on *Giardia* species found in lambs [23,31], calves, and cattle [32,33] in Türkiye, but there is no such study in buffaloes. In studies on buffaloes from other parts of the world, assemblage A [18,34,35] and assemblage E [18,20,23,27,28,29,30,31,32,33,36] were present. In this study, similar to previous studies [18,20,23,27,33,37,38], assemblage E was isolated from buffalo, but unlike other studies, assemblage B was also identified in these ruminants. To our knowledge, this study is the first time that assemblage B has been reported in buffaloes. The reasons for the difference between the studies include climate, immunity status, management conditions, and methods used.

A 9.3% prevalence of giardiasis in humans has been reported in the Van region, where this study was conducted, but the genotype was not reported [36]. It has been reported that *G. duodenalis* assemblage A3 was detected in a study performed on farm animals (sheep, goats, and cattle) in the Van region [30]. In molecular studies conducted in our country, the provinces in which *G. duodenalis* assemblage B is concentrated in humans have been reported [39,40,41,42,43]. In addition, Koloren et al. reported its presence in the waters of the Samsun and Giresun regions. Sursal et al. detected *G*. *duodenalis* assemblage A and B [44,45] in cats. The role of animals in the spread and epidemics of human giardiasis has been discussed. The primary source of *G. duodenalis* assemblage B is humans and is blamed for persistent infections in humans [46,47]. However, the fact that this assemblage was also isolated from domestic animals [48] indicates that this assemblage is anthropozoonotic. The isolation of *G. duodenalis* assemblage B in buffaloes in this study reveals that more research should be conducted on this subject.

## 5. Conclusions

Unfortunately, in today’s world, where people are dispersed everywhere, the incidence of zoonotic diseases is increasing. Giardiasis, one of these zoonotic diseases, is an important food- and waterborne disease that affects millions of people every year. The most critical factors in the spread of this parasite are its cyst form, which is highly resistant to environmental conditions, and cyst-containing feces that the reservoir hosts shed contaminate food and water sources, potentially causing epidemics. Ruminants are at the forefront of these reservoir management systems. Considering their numbers and the feces they excrete, ruminants play a critical role in the environment, agricultural products, and food and waters in which giardiasis is transmitted. Among these ruminants, buffalo also contribute to this spread. Therefore, determining the prevalence and assemblages of giardiasis in potential hosts by using molecular methods to prevent and manage this disease is crucial for human and animal health, that is, for the entire world’s health.

## Figures and Tables

**Figure 1 pathogens-12-00106-f001:**
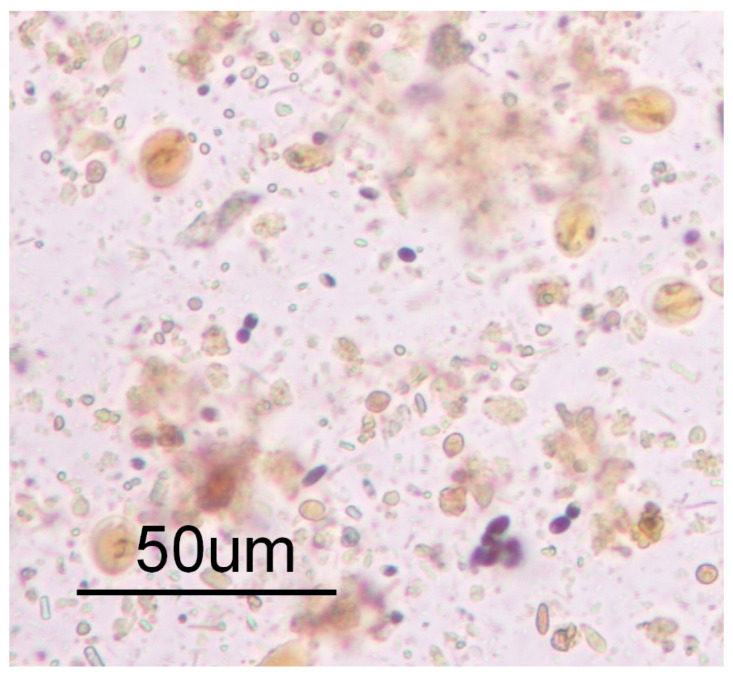
*Giardia* spp. cysts seen in a stool sample from a 4 month old calf with diarrhea (Sample 2) in the native Lugol examination.

**Figure 2 pathogens-12-00106-f002:**
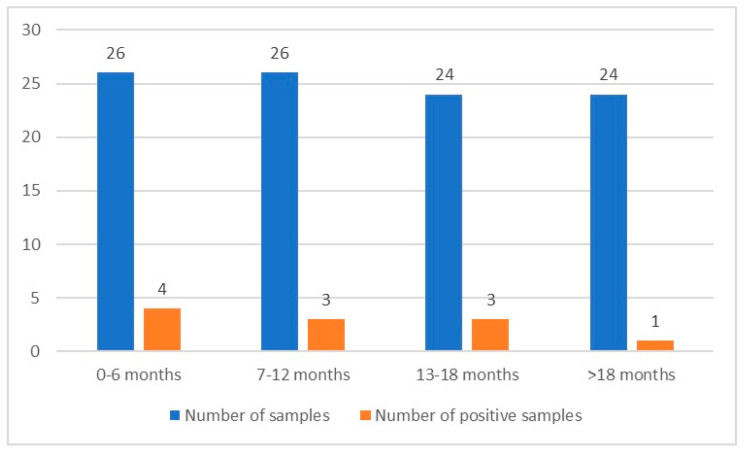
Numbers and age range of the samples.

**Figure 3 pathogens-12-00106-f003:**
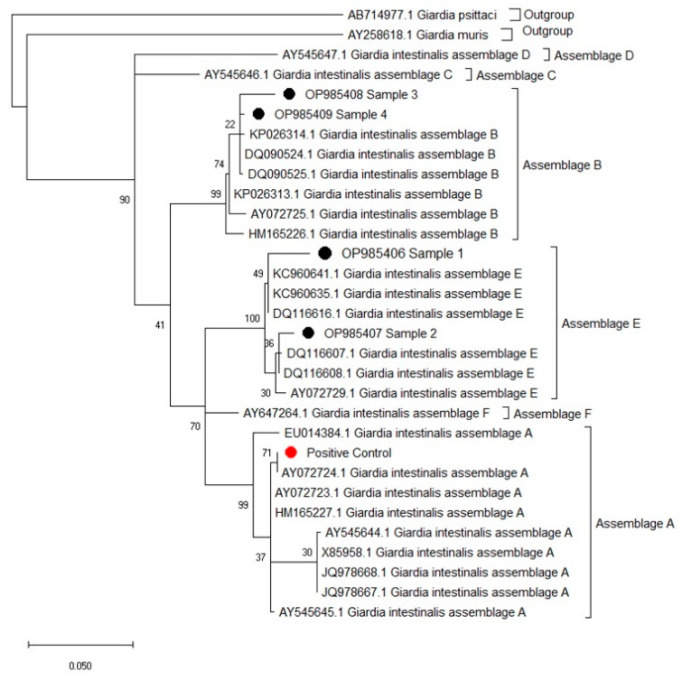
Phylogenetic relationships of *Giardia duodenalis* isolates via a maximum likelihood method analysis based on the β-giardin gene region. Numbers at the nodes represent the bootstrap values (1000 bootstraps). *Giardia psitacci* and *Giardia muris* were used as an outgroup. ●: Study samples; ●: Positive control.

**Table 1 pathogens-12-00106-t001:** Comparison results of the study samples generated using the NCBI Basic Local Alignment Search Tool.

Sequence Analyzed Samples and Accession Numbers	Age	Access Codes of the Most Similar Example and Similarity Rate	
		Assemblage B (MG736344)	Assemblage E (KT922248)
Sample 1 (OP985406)	4 months	93.95%	99.34%
Sample 2 (OP985407)	4 months	94.19%	99.57%
Sample 3 (OP985408)	9 months	98.85%	93.35%
Sample 4 (OP985409)	3 months	99.58%	94.80%

## Data Availability

The obtained nucleotide sequences were deposited in GenBank under the following accession numbers: OP985406, OP985407, OP985408, and OP985409.

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
