# Peer review of "The Investigation of Giardiasis (Foodborne and Waterborne Diseases) in Buffaloes in Van Region, Türkiye: First Molecular Report of Giardia duodenalis Assemblage B from Buffaloes"

_pathogens, 2023, doi:10.3390/pathogens12010106_

Round 1

Reviewer 1 Report

The manuscript reports a detection of Giardia duodenalis in Buffaloes fecal specimens.

However, there are many things needed to be improved, as below.

major points

1.Inroduction is too long. It is important to focus on why study Giardia duodenalis in Buffaloes in Turkey.

e.g. Lines 39-26: introduction need more focus on "foodborne and waterborne illnesses", and need citation of literture. 

Lines 64-71: logic confusing, please re-organize the sentences.

2.Results

Line 157: Could authors provide any photo from "native lugol examination"?

3.What is the relationship of 8 cyst-positive samples and PCR positives? particularly the sequenced ones? Does any buffaloes have  symptom?

4.Lines 165-167, Fig 1, table 1 and Fig 2, Lines 171-173, etc: samples of 5, 11,22 or samples BCDE, or samples 1-4, please unify them. Suggest not to use samples B/C/D/E, as it may confuss reader to Assemblage B/C/D/E.

5.Lines 165-167: only showing each sample 9x.xx% similar to only one Assemblage of B or E is not very helpful. Instead, each sample's similarities to each Assemblage is better.

6.Disscussion

Lines 214-228: I feel this part like result not discussion. And here are some aspects that the author could discuss more. 

e.g. How is Giardia duodenalis infection in Turkey, in human cases and other animals. Are they the same Assemblage or not?

What the new sequences may tell us, hybridization? SNP? 

minor points

Line 51: what is "intestinal flagella"?

Line 53: I don't think infection occurs "by direct contact with infected living things (humans or animals)". It occurs after being ingested.

Line 164: should be “β-giardin gene”.

Author Response

Corrections have been made according to your instructions and corrections have been made by  English editing system.

Reviewer 2 Report

The manuscript pathogen 2079681 has been reviewed. Giardia duodenalis assemblage B was found for the first time in the world in buffaloes. Because of the zoonotic properties of assemblage B this is an interesting finding, and the manuscript is reasonably well written with some inaccuracies. However, the data are limited and therefore may be a short communication is more appropriate?

 I have one major remark. They used nested PCR to amplify the bg fragment, which makes sense, because this is needed to get enough DNA for sequencing. But the nested PCR is prone to contamination. Therefore, my most important comment is that from the data provided by the authors it is difficult to excluded contamination from the positive control or otherwise. I would suggest to sequence also the positive control and indicate in the manuscript the source of the positive control (If it is from human or other primates it is very well possibly also assemblage B) and/or repeat the PCR on the sample without the positive control or use the assemblage E sample as positive control and re-sequence the samples which are expected to be assemblage B. Because assemblage B is not found before in buffalo, you have to be sure that it is no artefact.

Minor remarks:

Line 48-49. Include G. lamblia as syn. for G. duodenalis and G. intestinalis.

Line 51. Remove intestinal from flagella.

Line 53. Change “living things” into “animals” or “mammals”.

Line 56. Spp in Giardia spp. Must not be written in italic. Here and elsewhere.

Line 62. Do not start a sentence with an abbreviation, like G. duodenalis.

Line 71. Change “human health” into “human disease”?

Line 73-74. The statement that even 10 cysts can cause an infection misses a reference. This statement is seen in many papers, but I always wonder why you can’t get an infection with 1 or 2 cysts?

Line 81-85: Can it be useful to add the scientific name of the buffalo in order to avoid confusion?

Line 104-105. The age, sex and consistency were noted in the methods, but were not mentioned in the results. You can include these data in the supplementary data, see comments on table 1. If you do not give the results, you have to remove it from the method as well.

Line 122. Change “binding” into “annealing”        

Line 125. The reverse primer is not the primer as given by Lalle et al. The correct sequence is: 5′-CTCGACGAGCTTCGTGTT-3′. Hopefully, it is a typo and the correct primer is used for the experiments.

Line 142-146. Why were the indicated fragments used for comparison?

Line 158-164. Were the 8 positive samples  with native lugol methode also PCR positive? Maybe you can provide a table (in supplementary data?) which samples are positive in native lgol method and which samples were positive in PCR.  How can you know from the gel that the size is 511 bp? Change into “of the expected size”. What was wrong with the 7 positive samples that could not be sequenced? Was a product of unexpected size produced?

Line 165. Here and elsewhere. Abbreviate “Basic Local Alignment Tool “ into Blast. Change line 165 into “sample B was 99.78% identical to xxx (accession number) of assemblage E”. Also for the other samples. Please, submit the new sequences to Genbank and provide an accession number in the manuscript.

Line 173.  Change “present” into “found” or “identified”.

Figure 1. Why are not all the samples shown? Or at least the 11 positive samples. Now there are 3 (random?) samples shown. Why? The figure can be shown in the supplementary data.

Table 1. Give the sample numbers instead of imaginary codes B, C, D and E. In that way they can be linked to the raw data in the supplementary data (with age. sex, consisyency?). Change similarity rate into % identity.

Table 2. Can be moved to supplementary data.

Line 191 and elsewhere. Giardiasis is not with a capital.

Line 203 to 206. Difficult to read. Rephrase.

Lines 210 to 228. Here the differences are described at all the positions. However, that belongs to the results and not to the Discussion. If you want to give all these details, may be a table is more appropriate?

Line 210 and 214. Remove the statement that assemblage B infects humans. The authors have made that clear already several times.

Line 231. What is meant by “such environments”?

Line 239-241. Is Giardia really crucial for human health?

Author Response

(The authors gave the same response as above.)

Round 2

Reviewer 1 Report

The manuscript has been siginificantly improved. However, I can't find point-to-point response for the reviewers' comments.

In addition, I found Lines 171-173, 183, Table 1. are still using sample B for sample 1;  sample C for sample 2;  sample D for sample 3;  sample E for sample 4; 

Fig2 and legend still use names as samples 5, 11,22.  Suggest to  unify them as sample 1/2/3/4

Author Response

All required revisions have been made

Reviewer 2 Report

Some improvements are made. However, I still find it difficult to read while the experiment is quit straight forward and the figures and tables are inaccurate or not present!

Line 56 and elsewhere: Giardia has to be written in Italic.

Line 59. Change As Many as…. into As little as…

Line 62. Change “community genotypes” into assemblages. In other part also other names are used like assemblage genotypes (line 67). Change them into “assemblages”.

Line77. What does the author means by “specific animals”?

Line 85. There is nothing between the year 2021 and 2022. Must be 2021 and 2022?

Line 148. Change into: Bands of the expected size were obtained in 11 samples (11%).

Line 150. This remark about the hosts is out of place here.

Line 156-168. Rephrase. You can not say that 99.78% of the sequence of a sample belongs to a certain assemblage. Rephrase to : sample x was 99.78% identical with  (accession code)  from assemblage xx.

Line 159. Table 2 is not present, moved to supplementary data.

Line 169-192: a lot of text which for a great part can be found in the table and figure. Is all this text really needed? Especially where you talk about silent mutations. Leave that out, also from the discussion, because it is pure speculation.

Figure 1 and 2. More explanation in the legend is needed.

Table 1. I think not the whole table is visible

Figure 3. This is a clear figure with a good legend.

Lines 250-259. Pure speculation. If you want to say something about selection you need far more data.

Below are my remarks on the comments of the authors in blue

I have one major remark. They used nested PCR to amplify the bg fragment, which makes sense,

because this is needed to get enough DNA for sequencing. But the nested PCR is prone to

contamination. Therefore, my most important comment is that from the data provided by the authors it

is difficult to excluded contamination from the positive control or otherwise. I would suggest to

sequence also the positive control and indicate in the manuscript the source of the positive control (If it is from human or other primates it is very well possibly also assemblage B) and/or repeat the PCR on the sample without the positive control or use the assemblage E sample as positive control and re-sequence the samples which are expected to be assemblage B. Because assemblage B is not found before in buffalo, you have to be sure that it is no artefact.

In the PCR analysis, we used the A genotype sample isolated from the calves in our previous study as a positive control.

1. Ayan, A.; Ural, D.A.; Erdogan, H.; Kilinc, O.O.; Gültekin, M.; Ural, K. Prevalance and molecular characterization

of Giardia duodenalis in livestock in Van, Turkey. Int J Ecosyst Ecol Sci 2019, 9, 289-296.

OK, that is fine, but you have to give that information in the manuscript

Line 81-85: Can it be useful to add the scientific name of the buffalo in order to avoid confusion?

Added

The preferred genus name of buffalo is Bubalus, not Bos, I think.

Line 104-105. The age, sex and consistency were noted in the methods, but were not mentioned in the results. You can include these data in the supplementary data, see comments on table 1. If you do not give the results, you have to remove it from the method as well.

Table added

The samples taken in the study were divided into 0-6 months, 7-12 months, 13-18 and older than 18

months.

Since the number of buffaloes in our region is low, our sample number was low. We did not include

these results because we did not do statistics.

But we added it according to your instruction.

Table 1 seems only partly displayed. It is truncated somehow. And I see nothing of the negative animals, the sex of the animals (apart from line 150) and the consistency of the feces. Again, if you no not show the results, why mention it in the methods? Just one excel sheet with all 100 samples in rows and all the results (age, sex, consistency, lugol results, PCR results, assemblage) in columns would have made the results much more accessible.

Line 142-146. Why were the indicated fragments used for comparison?

The data in which access codes are presented for comparison were obtained from the NCBI Nucleotide database and samples in these access codes were selected to show their relationship to the Assemblege A, B, C, D, E, and F on the phylogenetic tree. In the selection of the data, the BLAST results of the study samples were taken into consideration first. In the dataset created with these data, Giardia psittaci (AB714977) and Giardia muris (AY258618) B-giardin gene sequences were determined as the outgroup. One sequence each was selected for Assemblege C, D and F with less host diversity, 9 sequences each for Assemblege A with more host diversity, and 6 sequences each for B and E. More sequences were selected for Assemblege E, which is important for buffaloes, and Assemblege A and B, which are zoonotic.

But there are thousands of bg sequences of assemblage B in Genbank, why did you pick these sequences? And you have to include  the arguments in the manuscript, not only in the reviewers reply.

Line 158-164. Were the 8 positive samples with native lugol methode also PCR positive? Maybe you can provide a table (in supplementary data?) which samples are positive in native lgol method and which samples were positive in PCR. How can you know from the gel that the size is 511 bp? Change into “of the expected size”. What was wrong with the 7 positive samples that could not be sequenced? Was a product of unexpected size produced?

All samples were sequenced by sanger dideoxy sequencing; However, it is not given in the article in order not to give an incorrect result due to the early termination of sequencing in some of the 7 positive samples whose results were not given. You can see the sequence graphs of a few examples with no results below and understand why we didn't include them in the article

You do not have to show the non-informative sequence data, only mention in the manuscript that sequences from those samples were of insufficient quality.

Figure 1. Why are not all the samples shown? Or at least the 11 positive samples. Now there are 3 (random?) samples shown. Why? The figure can be shown in the supplementary data.

We put our clearest photo in the article..

You have to mention than that the figure is a selection of the clearest bands.  The picture of the gel with the PCR products now in the supplementary data cannot be understood without any explanation. All the data in the supplementary data have to be ordered much better, with a legend so that the reader can understand. And you can leave out figure 2 than from the manuscript.

Table 1. Give the sample numbers instead of imaginary codes B, C, D and E. In that way they can be linked to the raw data in the supplementary data (with age. sex, consisyency?). Change similarity rate into % identity.

This is how we submitted it for sequence analysis, so we can't remove it. But we added the access codes and the number

Now we have the samples indicated in table 1  with a number+digit, in figure 2 with other digits, in figure 3 with a letter + accession code and in the text lines 156-168 with digits +accession codes and in lines 169-173 with  letters again. Very inconsistent. Why not use the sample numbers throughout the manuscript and link those numbers once with the accession numbers.

Author Response

But there are thousands of bg sequences of assemblage B in Genbank, why did you pick these sequences? And you have to include  the arguments in the manuscript, not only in the reviewers reply.

answer: Of course there are hundreds of bg sequences in genbank; however, many different primer sets have also been designed for the diagnosis of giardia by targeting the bg gene, and sequences covering different regions of the gene can also be found in the genbank database. The main reason we prefer these access codes is that the primer sets we use in the PCR process are compatible with these regions. In addition, assemblage B and E datasets were created according to the BLAST results of the study samples, and care was taken that the datasets of other assemblages included the sequence of the PCR product we produced.

All requested revisions have been made.